

# Genome-wide identification and characterization of the fibrillin gene family in *Triticum aestivum*

Yaoyao Jiang[1,*], Haichao Hu[2,*], Yuhua Ma[3] and Junliang Zhou[3]

[1] School of Forestry and Biotechnology, Zhejiang Agriculture and Forestry University, Hangzhou, China
[2] College of Agriculture and Food Science, Zhejiang Agriculture and Forestry University, Hangzhou, China
[3] Guizhou Institute of Pomological Sciences, Guizhou Academy of Agricultural Sciences, Guiyan, China
[*] These authors contributed equally to this work.

## ABSTRACT

**Background**. The fibrillin (*FBN*) gene family is highly conserved and widely distributed in the photosynthetic organs of plants. Members of this gene family are involved in the growth and development of plants and their response to biotic and abiotic stresses. Wheat (*Triticum aestivum)*, an important food crop, has a complex genetic background and little progress has occurred in the understanding of its molecular mechanisms.
**Methods**. In this study, we identified 26 *FBN* genes in the whole genome of *T. aestivum* through bioinformatic tools and biotechnological means. These genes were divided into 11 subgroups and were distributed on 11 chromosomes of *T. aestivum*. Interestingly, most of the *TaFBN* genes were located on the chromosomes 2A, 2B and 2D. The gene structure of each subgroup of gene family members and the position and number of motifs were highly similar.
**Results**. The evolutionary analysis results indicated that the affinities of *FBNs* in monocots were closer together. The tissue-specific analysis revealed that *TaFBN* genes were expressed in different tissues and developmental stages. In addition, some *TaFBNs* were involved in one or more biotic and abiotic stress responses. These results provide a basis for further study of the biological function of *FBNs*.

## INTRODUCTION

Fibrillins (FBNs) are named after fibrils because these proteins were first detected in fibrils in the chromoplasts of dog rose (*Rosa rugosa*) and bell pepper (*Capsicum annuum*) fruit (*Newman, Hadjeb & Price, 1989*; *Deruère et al., 1994*; *Kim, Lee & Kim, 2015*). Since then, FBN proteins have been found in different organelles, including the plastoglobules (PGs) in the chloroplasts and algal eyespots. Therefore, members of the FBN protein family have been given many different names, including the plastid-lipid associated protein (PAP), the plastoglobule (PGL), the chloroplastic drought-induced stress protein of 34 kDa (CDSP 34), and the chromoplast-specific carotenoid-associated protein (ChrC) (*Pozueta-Romero et al., 1997*; *Ting et al., 1998*; *Kim, Lee & Kim, 2015*). Fibrillins are located in the photosynthetic organs of cyanobacteria and some higher plants (*Kim, Lee & Kim, 2015*; *Kim et al., 2017*).

Corresponding author
Junliang Zhou, gsszjl2008@163.com

*Lundquist et al. (2012)* identified 14 *FBN* genes in *Arabidopsis* by proteomic analysis, 50% of which were in the PGs, while others are mainly distributed in the stroma and thylakoid membranes (*Lundquist et al., 2012*; *Kim, Lee & Kim, 2015*).

So far, the FBN protein family is mainly composed of 12 subfamilies; 11 of these have been found in higher plants and one has been identified in algae (*Lohscheider & Río Bártulos, 2016*; *Kim et al., 2017*). The members of these subfamilies were found to have similar hydrophobic structures; however, the biophysical properties of these proteins are quite diverse, including proteins with molecular weights of 20–42 kDa and isoelectric point (pI) values of 4–9 (*Vidi et al., 2006*; *Lundquist et al., 2012*). These findings suggest that each FBN protein may have specific biological functions. In *Arabidopsis thaliana*, FBN proteins contain a conserved hydrophobic domain (lipocalin motif 1) in the N-terminus and amino acid residues near the C-terminus, including aspartic acid (*Singh et al., 2010*). Furthermore, *Lohscheider & Río Bártulos (2016)* predicted that the three-dimensional structure of FBNs is similar to that of lipocalin, with the ability to bind and transport small hydrophobic molecules (*Lohscheider & Río Bártulos, 2016*), which suggests that the FBN family may have similar biological functions (*Singh et al., 2010*; *Francesc & Albert, 2015*; *Kim, Lee & Kim, 2015*).

FBN proteins have a variety of important biological functions, such as participating in photosynthesis, the formation of lipoprotein structures, and responses to abiotic and biotic stresses (*Kim, Lee & Kim, 2015*). Initially, researchers found that FBNs are located on the outer surface of red pepper chromoplast fibrils by Immunogold electron microscopy (*Deruère et al., 1994*). Furthermore, fibril-like structures can be reconstituted *in vitro* from a mixture of FBN protein, lipids, and bicyclic carotenoids (*Deruère et al., 1994*; *Kim, Lee & Kim, 2015*). Compared to wild-type plants, RNAi-transgenic tomato plants with suppressed *LeChrC* (FBN1) accumulate 30% fewer carotenoids (*Leitner-Dagan et al., 2006*; *Singh et al., 2010*). In addition, when the *FBN5* gene was deleted in *Arabidopsis thaliana* and rice, mutant plants were more sensitive to light stress, and the levels of PQ-9 and PC-8 in the leaves were reduced (*Kim et al., 2017*). These results suggest that FBNs can regulate the formation of chromoplast fibrils and the accumulation of carotenoids. In addition to structural roles, fibrillin gene expression is also regulated by numerous abiotic and biotic stresses, especially oxidative stress (*Youssef et al., 2010*). For example, the expression of the *Chrc* (*FBN1*) is induced in cucumber leaves infected with *Sphaerotheca fuliginea* (*Leitner-Dagan et al., 2006*). Similar results were seen for tomato plants infected with the fungus *Botrytis cinerea*. However, the expression patterns of fibrillins are varied and complex during abiotic stress such as heat, cold, drought, high light and wounding treatment (*Pruvot et al., 1996*; *Kuntz et al., 1998*; *Langenkämper et al., 2001*; *Leitner-Dagan et al., 2006*; *Simkin et al., 2008*). *AtFBN1a* expression was induced and *AtFBN2* expression was repressed when subjected to drought or cold treatment (*Laizet et al., 2004*). High levels expression of fibrillin (*FBN1*) were observed in potato plants during water stress (*Lee et al., 2007*). Similarly, the mutants of *pgl1* and *pgl2* were more sensitive to high light stress than was the wild-type in *Synechocystis* sp (*Cunningham et al., 2010*). Moreover, when *LeChrC* (*FBN1*), *FBI4* and *AtFBN4* were knocked down in tomato, apple, and *Arabidopsis*, the mutant plants were more susceptible to the phytopathogenic fungus *Botrytis cinerea* and pathogenic bacteria

*Erwinia amylovora* and *Pseudomonas syringae* pv. *tomato*, respectively (*Cooper et al., 2003*; *Leitner-Dagan et al., 2006*; *Singh et al., 2010*). Meanwhile, *FBN* gene expression is regulated by hormones, including gibberellic acid (GA), jasmonate, and abscisic acid, during plant growth and developmental stages, as well as when plants are subjected to stresses (*Yang et al., 2006*; *Youssef et al., 2010*; *Kim et al., 2017*). The accumulation of FBN proteins in the tomato *flacca* mutant plant was decreased, which was defective in ABA biosynthesis, when subjected to drought stress. The level of FBN protein can be induced by abscisic acid treatment (*Gillet et al., 2001*). Moreover, *FBN1* and *FBN2* proteins are involved in the jasmonate biosynthesis pathway in response to light and cold stress (*Youssef et al., 2010*). By contrast, *FBN1* mRNA and protein levels declined in red pepper fruit when treated with gibberellic acid (*Deruère et al., 1994*).

Wheat (*Triticum aestivum* L.) is an important food crop that is widely grown around the world. Approximately 40% of the global population depends on *T. aestivum* as their staple food (*Paux et al., 2008*; *Han et al., 2019*). Common *T. aestivum* is a heterogenous hexaploid containing A, B, and D genomes; therefore, the genome information is large and complex (*Ling et al., 2013*; *Glover et al., 2015*; *Han et al., 2019*). Moreover, owing to the complex genetic background of *T. aestivum*, only some genes regulating important agronomic and disease-related traits were reported. Therefore, the study of *T. aestivum* functional genomics is lagging far behind that of rice and corn. In recent years, high-quality wheat genome sequencing has been completed (*International Wheat Genome Sequencing Consortium et al., 2018*); this will play an important role in elucidating the molecular mechanisms involved in growth and development, resistance, and high yield (*Pradhan et al., 2019*; *Rahimi et al., 2019*).

Although there is increasing evidence that *FBN*s play major roles in photosynthetic organisms, to date, they have been identified and characterized from only a few plant species. In addition, there are few studies on the function of FBN genes in wheat. The identification and functional characterization of the *FBN* family in *T. aestivum* will contribute to elucidating the stress response mechanisms. In this study, we performed a genome-wide survey using the reported FBN protein sequences in the *T. aestivum* database. We identified 26 *FBN* genes in *T. aestivum* and used bioinformatic methods to analyze their biophysical properties, including gene structures and conserved motifs, as well as the chromosome distribution of the *FBN* genes. In addition, we analyzed the expression profiles of *TaFBN* genes in different tissues, at different developmental stages, and in response to abiotic and biotic stresses using the *T. aestivum* expression database. These results may provide a basis for studying the biological function of the *FBN* gene in different growth and development stages of *T. aestivum*.

## MATERIALS & METHODS

### Plant material cultivation and treatments

The common *T. aestivum* cultivar "Chinese spring" was used in this study. *Triticum aestivum* seeds were sterilized with 1% NaOCl for 15 min, rinsed thoroughly with distilled water five times, and soaked in distilled water overnight at room temperature (18 °C). The

seeds were transferred to filter paper and germinated for three days. The seedlings were cultured in a nutrient solution and grown in a growth chamber with 16 h light (22 °C), 8 h dark (18 °C), and 50% humidity. The nutrient solution was replaced every three days at the growth stage. At 21 days old, the seedlings were treated with 20% (m/V) PEG 6000 (Sigma-Aldrich, St. Louis, MO, USA) for 6 h. Untreated seedlings were used as a control, and each treatment contained three independent biological replicates. The roots, shoots, and leaves were collected separately for further analysis at 1 h and 6 h after treatment.

## Identification of *TaFBN* genes

We used the protein sequences of *Arabidopsis thaliana FBN* (*AtFBN*) and *Oryza sativa* FBN (*OsFBN*) genes as queries to perform a BLAST (*E*-value le$^{-10}$) search against the *T. aestivum* genome database (genome assembly from IWGSC; http://ensembl.gramene.org/). We obtained a dataset of *TaFBN* sequences and filtered out the redundant sequences. The protein sequences of the *AtFBN* and *OsFBN* genes were downloaded from the *Arabidopsis* Information Resource database (https://www.arabidopsis.org/) and the Rice Annotation Project database (https://rapdb.dna.affrc.go.jp/). Since a typical FBN protein is reported to contain a conserved PAP_fibrillin domain (PF04755), the online tools SMART (http://smart.embl-heidelberg.de/) and InterProScan (http://www.ebi.ac.uk/interpro/) were used to predict the functional domains of the potential TaFBN proteins. To verify our results, all of the proteins were compared to the PAP_fibrillin domain using the HMMER 3.0 program, with the default *E*-value (*E*-value < 10$^{-3}$). Proteins without the PAP_fibrillin domain were removed. The biophysical properties of the final TaFBN proteins were calculated using the ExPASy ProtParam tool (https://web.expasy.org), including the theoretical values of pI, relative molecular mass, and the grand average of hydrophobicity (GRAVY). The subcellular localization of *TaFBN*s was analyzed using ProComp (http://linux1.softberry.com) and WoLF PSORT II (https://www.genscript.com/wolf-psort.html). In addition, the signal peptide and chloroplast transit peptides of the *TaFBN* genes were predicted using the SignalP 4.1 server (http://www.cbs.dtu.dk/services/SignalP-4.1/) and ChloroP 1.1 server (http://www.cbs.dtu.dk/services/ChloroP/).

## Multiple sequence alignments and phylogenetic analysis

Full-length protein sequences of the *FBN* gene family members identified in 13 plant species, including eight monocotyledon species and five dicotyledon species, were downloaded from the NCBI database (https://www.ncbi.nlm.nih.gov/), the Ensembl Plants database (genome assembly from IWGSC; http://ensembl.gramene.org/), and the Phytozome v12.1 database (https://phytozome.jgi.doe.gov/pz/portal.html). The full-length protein sequences of these *FBN* genes were aligned using MAFFT software (https://mafft.cbrc.jp/alignment/server/). Based on FASTA files, a neighbor-joining phylogenetic tree was constructed using Molecular Evolutionary Genetics Analysis (MEGA) version 7.0 software with 1000 bootstrap replicates. The phylogenetic tree we constructed in this study belongs to the phylogram tree. The phylogram tree is a branching diagram assumed to be an estimate of a phylogeny, branch lengths are proportional to the amount of inferred evolutionary change. Low support nodes (<50) was collapsed and high support nodes (>99) used a symbol (*) in phylogenetic tree.

### Analysis of gene structures and conserved motifs

To investigate the structure of *TaFBN* genes, we used the Gene Structure Display Server 2.0 database (http://gsds.cbi.pku.edu.cn/) to analyze the distribution of exons and introns in *TaFBN* genes. Conserved motifs were predicted using the Multiple EM for Motif Elicitation (MEME) database (http://alternate.meme-suite.org/); the number of motifs was set to 10 and the motif width was set to 6–50.

### Analysis of the *cis*-regulatory element of *FBN* gene promoters

In this study, 2,000-bp sequences upstream of the translational start sites of the *TaFBN* genes were set as promoter sequences. PlantCARE software (http://bioinformatics.psb.ugent.be/webtools/plant care/html/) was used to predict the *cis*-regulatory elements based on these promoter sequences. The distribution of *cis*-regulatory elements in the promoter of the *TaFBN* gene was displayed using TBtools software (https://github.com/CJ-Chen/TBtools) (*Chen et al., 2018*).

### Analysis of *TaFBN* gene expression patterns

The expression profile data used in this study were obtained via the Wheat Expression Browser database (http://www.wheat-expression.com/) (*Philippa, Ricardo & Cristobal, 2016*; *Ramírez-González et al., 2018*). We searched for *FBN* genes on the website using the gene ID as query terms. The expression of *TaFBN*s in different tissues, at different developmental stages, and under different abiotic and biotic stress conditions (including drought, cold, heat, and stripe rust) were analyzed. The results were presented as heatmaps, with different colors representing the absolute signal values. The color scale of the heatmap was given in $\log_2$ ratio values. The cultivar used in the gene expression profiles analysis was "Chinese spring".

### Total RNA isolation and real-time PCR analysis

Total RNA from different tissues was extracted using TRIzol Reagent (Invitrogen). The total RNA was treated with RNase-free DNase I for 15 min to remove the remaining genomic DNA. First-strand cDNA was synthesized according to the manufacturer's instructions (TOYOBO, Kita-ku, Osaka, Japan), diluted 20 times, and used as a template for quantitative real-time PCR (qRT-PCR), which was performed using AceQ qPCR SYBR Green Master Mix (Vazyme, Nanjing, China). For an endogenous control, we used the *T. aestivum actin* gene (AB181991). At least three biological replicates, with three technical replicates each, were used for each treatment. Relative expression levels were calculated using the comparative $2^{-\Delta\Delta Ct}$ method (*Willems, Leyns & Vandesompele, 2008*). The *TaFBN* primers used for qRT-PCR are listed in Table S1.

## RESULTS

### Identification and characterization of *FBN* genes in *T. aestivum*

In this study, a total of 26 FBN genes were identified in *T. aestivum*, which we named *TaFBN-A1*–*TaFBN-D10* according to their genome location (Table 1). The *TaFBN* characteristics, including the chromosomal position, intron number, gene length, number

of amino acids, molecular mass, CDS, subcellular localization, signal peptide, and instability index, are listed in Table 1. As shown in Table 1, the TaFBN protein sequences ranged from 219 to 402 amino acids and the molecular weights ranged from 23.75 to 43.59 kDa. The prediction of subcellular locations indicated that 18 TaFBNs were located in the chloroplasts and eight were located extracellularly. At present, GRAVY values are an important index of measuring protein hydrophobicity. The GRAVY values of most TaFBN proteins, except *TaFBN-A1*, *TaFBN-B1*, and *TaFBN-B6*, were negative, suggesting that they are hydrophilic. Meanwhile, the prediction results showed that no signal peptides were found in any TaFBN proteins, but all TaFBN proteins contain chloroplast transit peptides.

### Gene structure analysis of *TaFBN* genes

To gain insight into the evolution of the *TaFBN* gene family, a diagram of the *TaFBN* exon-intron gene structure was constructed based on the cDNA and genomic DNA sequence information (Text S1) using the Gene Structure Display Server (Fig. 1B). A neighbor-joining phylogenetic tree was also constructed to explore the evolutionary relationship and the phylogenetic classification of the *FBN* genes in wheat. Gene structure analyses indicated that homologous genes had similar exon-intron distribution patterns (Fig. 1B). However, the number of introns in different *TaFBN* gene family members varied greatly (ranging from 2 to 10 introns), while there was almost no difference between members of the same subfamily. We found that all *TaFBN* genes contained a conserved PAP_FBN domain (PF04755), and the distribution of the domains was consistent with the genetic homology (Fig. 1C). These results suggested that members of the same subfamily may have similar biological functions. In addition, we used the MEME online tool to analyze the conserved motifs of the *TaFBN* genes; the results showed that all *TaFBN* members contained five to nine conserved motifs (Fig. 2). The logo representation of the 10 conserved motifs identified for the proteins encoded by the *TaFBN* genes is described in Fig. S1. Figure 2 showed that motif 1, motif 2, motif 3, motif 4, and motif 5 were highly conserved and widely distributed in all TaFBN proteins. The motif/domain analysis revealed that motif 1 contained conserved amino acid residues in the C-terminal and motif 3 contained a conserved lipocalin motif (Fig. S2). The types and distribution of the conserved motifs may be the reason for the functional diversity of the *TaFBNs*.

### Phylogenetic and evolutionary analysis of *TaFBN*

An unrooted phylogenetic tree was constructed for 183 *FBN* genes from eight monocotyledon species (with 26 *FBNs* from *T. aestivum*, nine from *Oryza sativa*, 11 from *Zea mays*, 10 from *Sorghum bicolor*, nine from *Panicum hallii*, 20 from *Panicum virgatum*, 10 from *Setaria italica*, and eight from *Hordeum vulgare*) and five dicotyledon species (with 14 *FBNs* from *A. thaliana*, 12 from *Brassica oleracea* var. *capitata*, 11 from *Nicotiana tabacum*, 21 from *Glycine max*, and 22 from *Coffea arabica*) to study the evolutionary relationships of the *TaFBN* members (Fig. 3). Based on the *FBN* gene characteristics of *A. thaliana*, these *FBN* genes can be classified into 11 subfamilies (Group 1 to Group 11). Interestingly, the members of the *TaFBNs* were identified into nine subfamilies, each

Jiang et al. (2020), *PeerJ*, DOI 10.7717/peerj.9225

Peer

**Table 1** *Fibrillin* (FBN) gene family in *Triticum aestivum.*

| Gene name | Sequence ID | Chromosome | Genomic position | Intron number | Gene length (aa) | Molecular weight (kDa) | pI | Predicted pfam domain | Subcellular prediction by PC | Grand average of hydropathicity | Signal peptides | Chloroplast transit peptides |
|---|---|---|---|---|---|---|---|---|---|---|---|---|
| TaFBN-A1 | TraesCS5A02G164600.1 | Chr5A | 353189098-353192310 | 2 | 314 | 33.06 | 7.77 | PAP_fibrillin | Chloroplast | 0.039 | NA | Y |
| TaFBN-B1 | TraesCS5B02G162100.1 | Chr5B | 299020240-299020330 | 2 | 312 | 32.94 | 7.77 | PAP_fibrillin | Chloroplast | 0.056 | NA | Y |
| TaFBN-A2 | TraesCS1A02G193500.1 | Chr1A | 350749390-350752293 | 2 | 360 | 38.27 | 4.79 | PAP_fibrillin | Chloroplast | −0.261 | NA | Y |
| TaFBN-B2 | TraesCS1B02G208500.1 | Chr1B | 378397002-378399661 | 2 | 360 | 38.33 | 4.83 | PAP_fibrillin | Chloroplast | −0.294 | NA | Y |
| TaFBN-D2 | TraesCS1D02G197400.1 | Chr1D | 278512124-278514657 | 2 | 360 | 38.28 | 4.79 | PAP_fibrillin | Chloroplast | −0.253 | NA | Y |
| TaFBN-A3 | TraesCS4A02G272000.1 | Chr4A | 583754471-583757208 | 5 | 261 | 28.59 | 9.34 | PAP_fibrillin | Extracellular | −0.33 | NA | Y |
| TaFBN-B3 | TraesCS4B02G042000.1 | Chr4B | 28717109-28719740 | 5 | 260 | 28.48 | 9.61 | PAP_fibrillin | Chloroplast | −0.318 | NA | Y |
| TaFBN-D3 | TraesCS4D02G039200.1 | Chr4D | 16799419-16802059 | 5 | 261 | 28.55 | 9.21 | PAP_fibrillin | Chloroplast | −0.325 | NA | Y |
| TaFBN-A4 | TraesCS2A02G145900.1 | Chr2A | 90688741-90690297 | 3 | 275 | 28.99 | 8.95 | PAP_fibrillin | Chloroplast | −0.244 | NA | Y |
| TaFBN-B4 | TraesCS2B02G171300.1 | Chr2B | 144596063-144597581 | 3 | 276 | 29.35 | 9.51 | PAP_fibrillin | Chloroplast | −0.267 | NA | Y |
| TaFBN-D4 | TraesCS2D02G150500.1 | Chr2D | 93046450-93048107 | 3 | 277 | 29.41 | 9.51 | PAP_fibrillin | Chloroplast | −0.277 | NA | Y |
| TaFBN-A5 | TraesCS2A02G300200.1 | Chr2A | 515959001-515961447 | 6 | 262 | 28.96 | 9.16 | PAP_fibrillin | Chloroplast | −0.213 | NA | Y |
| TaFBN-B5 | TraesCS2B02G316500.1 | Chr2B | 451833336-451836114 | 6 | 260 | 28.67 | 9.28 | PAP_fibrillin | Chloroplast | −0.178 | NA | Y |
| TaFBN-D5 | TraesCS2D02G298100.1 | Chr2D | 380429694-380432365 | 6 | 256 | 28.43 | 9.36 | PAP_fibrillin | Chloroplast | −0.2 | NA | Y |
| TaFBN-A6 | TraesCS2A02G431000.1 | Chr2A | 684246511-684248296 | 3 | 219 | 23.78 | 8.8 | PAP_fibrillin | Extracellular | −0.044 | NA | Y |
| TaFBN-B6 | TraesCS2B02G452300.1 | Chr2B | 646214215-646215789 | 3 | 219 | 23.75 | 8.73 | PAP_fibrillin | Chloroplast | 0.003 | NA | Y |
| TaFBN-D6 | TraesCS2D02G428800.1 | Chr2D | 540824383-540826044 | 3 | 219 | 23.82 | 8.74 | PAP_fibrillin | Chloroplast | −0.031 | NA | Y |
| TaFBN-A7 | TraesCS2A02G487900.1 | Chr2A | 722519297-722522892 | 6 | 297 | 32.55 | 5.73 | PAP_fibrillin | Chloroplast | −0.231 | NA | Y |
| TaFBN-B7 | TraesCS2B02G515500.1 | Chr2B | 710281451-710285064 | 6 | 281 | 30.92 | 6.06 | PAP_fibrillin | Chloroplast | −0.247 | NA | Y |
| TaFBN-D7 | TraesCS2D02G488200.1 | Chr2D | 587697352-587701032 | 6 | 293 | 32.03 | 5.35 | PAP_fibrillin | Chloroplast | −0.171 | NA | Y |
| TaFBN-A9 | TraesCS2A02G413700.1 | Chr2A | 670791911-670794116 | 2 | 222 | 24.21 | 6.74 | PAP_fibrillin | Extracellular | −0.106 | NA | Y |
| TaFBN-B9 | TraesCS2B02G432500.1 | Chr2B | 621664679-621666987 | 2 | 222 | 24.27 | 7.9 | PAP_fibrillin | Extracellular | −0.123 | NA | Y |
| TaFBN-D9 | TraesCS2D02G410900.1 | Chr2D | 525935293-525937697 | 2 | 222 | 24.24 | 7.9 | PAP_fibrillin | Extracellular | −0.136 | NA | Y |
| TaFBN-A10 | TraesCS2A02G434800.1 | Chr2A | 686874975-686880122 | 10 | 401 | 43.44 | 9.31 | PAP_fibrillin | Extracellular | −0.16 | NA | Y |
| TaFBN-B10 | TraesCS2B02G455900.1 | Chr2B | 650399573-650405456 | 10 | 402 | 43.59 | 9.19 | PAP_fibrillin | Extracellular | −0.182 | NA | Y |
| TaFBN-D10 | TraesCS2D02G432600.1 | Chr2D | 544665056-544670351 | 10 | 398 | 43.12 | 9.11 | PAP_fibrillin | Extracellular | −0.152 | NA | Y |
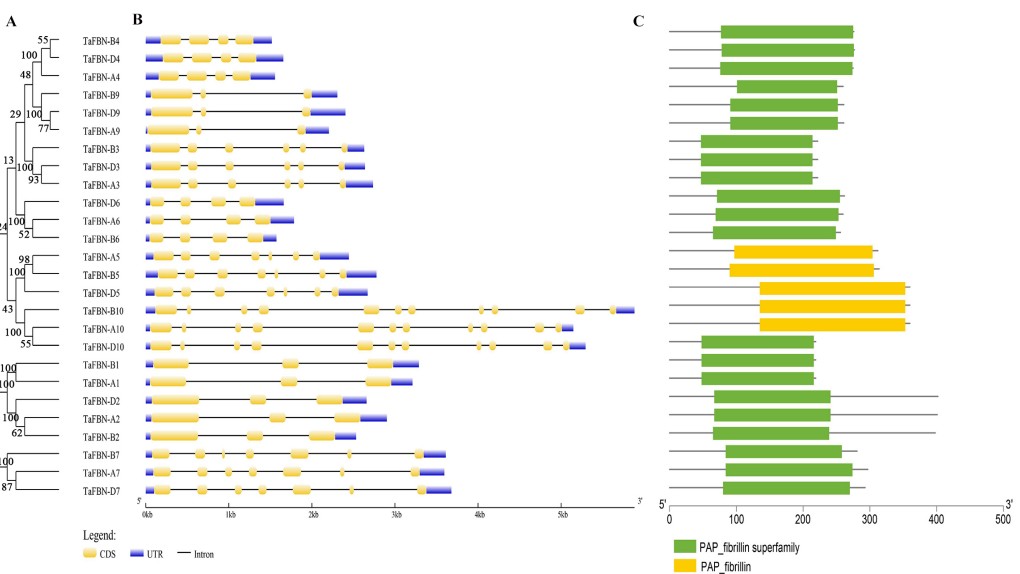

**Figure 1  Phylogenetic relationships, gene structure and functional domain analysis of the TaFBN proteins.** (A) A phylogenetic tree using the neighbor-joining method in MEGA7, with bootstrap values of 1,000, was constructed to determine whether the exon-intron distribution patterns correlated with the phylogenetic classification of *TaFBN* (the same phylogenetic tree is also shown in Figs. 2 and 4). (B) The coding sequences (CDS) of exons are represented by yellow boxes, the introns are represented by lines, and the untranslated regions (UTRs) are indicated by blue boxes. (C) The conserved domains of the TaFBN proteins were identified using the Conserved Domain Database (CDD) of NCBI against the Pfam v30.0 database (https://www.ncbi.nlm.nih.gov/Structure/cdd/wrpsb.cgi).

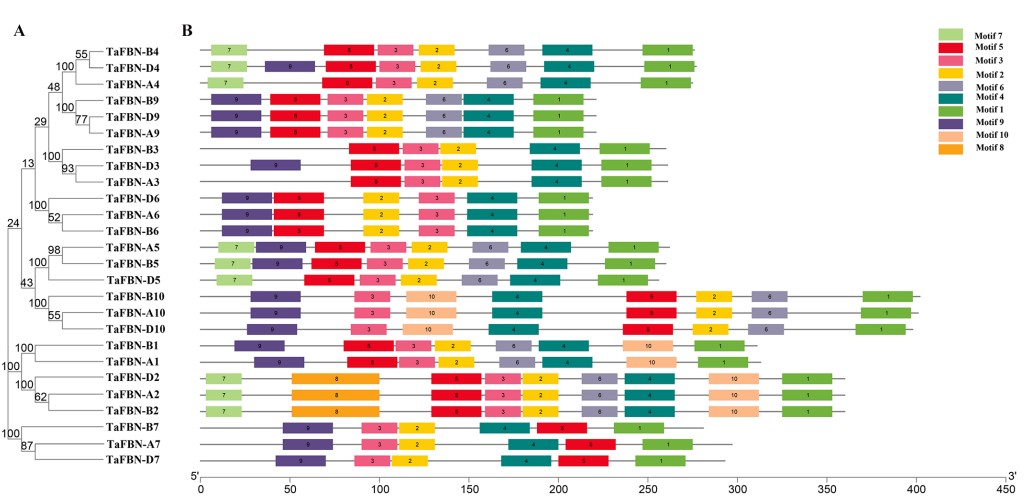

**Figure 2  The motif distribution of wheat FBN proteins.** (A) The phylogenetic tree of *TaFBN* genes was constructed using the neighbor-joining method in MEGA7, with bootstrap values of 1,000. (B) The conserved motifs were predicted using Multiple Em for Motif Elicitation (MEME) (http://alternate.meme-suite.org/). The box length indicates the number of amino acids in the motif.

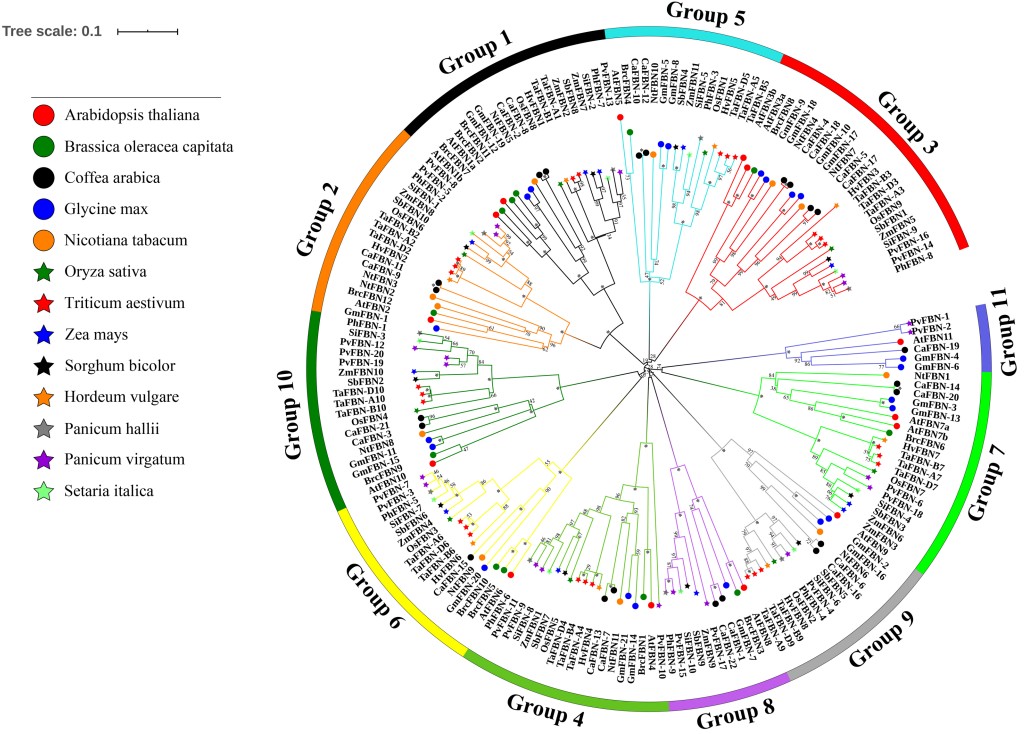

**Figure 3** **Phylogenetic analysis among 183 FBN proteins in different species.** The FBN proteins are clustered *Triticum aestivum*, *Oryza sativa*, *Sorghum bicolor*, *Zea mays*, *Panicum hallii*, *Panicum virgatum*, *Setaria italica*, *Hordeum vulgare*, *Arabidopsis thaliana*, and *Brassica oler*. The phylogenetic tree was inferred using the neighbor-joining method in MEGA7, with bootstrap values of 1,000. Bootstrap values are indicated at the nodes, where asterisks indicate values >99%. The low support nodes (<50) were collapsed and the high support nodes (>99) were represented by the symbol * in the phylogenetic tree.

subfamily containing two or three *FBN* genes. The analysis also revealed that the *FBN* genes in monocots (i.e., *T. aestivum*, *O. sativa*, *Z. mays*, *P. hallii*, and *S. bicolor*) were more closely related than those of the dicots (i.e., *A. thaliana*, *B. oleracea* var. *capitata*, and *N. tabacum*).

## Analysis of *TaFBN cis*-regulatory elements

To further identify the *cis*-regulatory elements located upstream of the *TaFBN* genes, 2000-bp sequences upstream from the translational start sites of putative *TaFBN* gene families were analyzed using the PlantCARE tool. As shown in Fig. 4, many *cis*-regulatory elements were identified in the promoters of the *TaFBN* genes. These *cis*-regulatory elements can be divided into three types: hormone response elements, stress response-related elements, and light response-related elements. The hormone response elements, including the methyl jasmonate (MeJA)-responsive, abscisic acid-responsive, gibberellin-responsive, salicylic acid-responsive, and auxin-responsive elements, were widely distributed in promoters of the *TaFBN*s. The responses to abiotic stress were the light response-related, low temperature response-related, and drought stress-related response elements. These results suggested that *TaFBN* genes may be involved in photosynthesis, stress responses, and maintaining

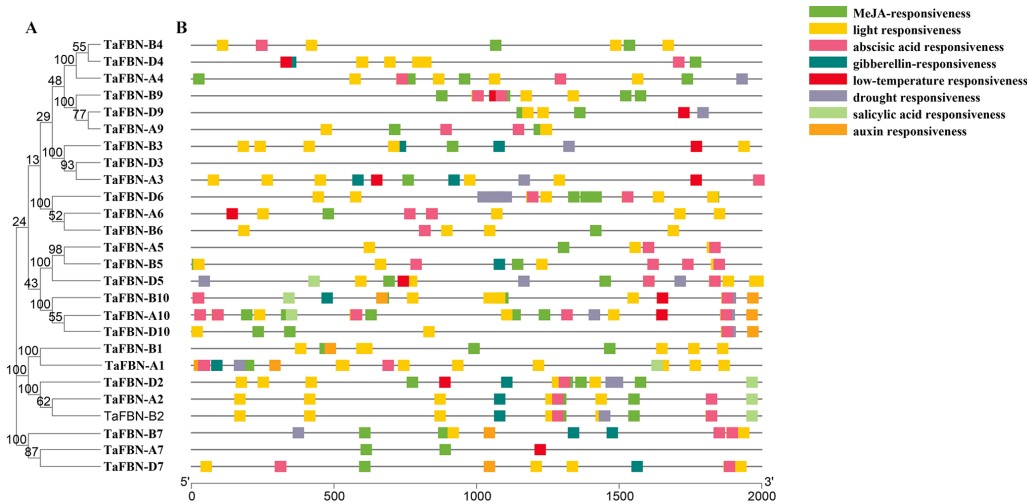

**Figure 4** **Predicted cis-regulatory elements in the *TaFBN* gene promoters.** (A) A phylogenetic tree inferred using the neighbor-joining method in MEGA7, with bootstrap values of 1,000, was constructed to determine whether the exon-intron distribution patterns correlated with the phylogenetic classification of *TaFBN*. (B) The promoter sequences (2,000 bp) upstream of genes were chosen for cis-regulatory element analysis using the PlantCARE online tool (http://www.dna.affrc.go.jp/PLACE/). Each color indicates a cis-regulatory element.

the hormone balance in plants, thereby improving the chances for organisms to escape or better cope with the damaging effects of adverse environmental conditions.

## Tissue specific expression patterns of *TaFBN*s at different developmental stages

To explore the tissue-specific expression patterns of *TaFBN* genes at different growth and developmental stages in *T. aestivum*, publicly available expression data sets for the 26 *TaFBN*s were analyzed. The transcription levels in various *T. aestivum* tissues, including the root, shoot, anther, spikelet, and leaf, were examined. Most of the *TaFBN* genes were detected in at least two or more different tissues. The results suggested that *TaFBN* genes may be widely expressed in wheat tissues (Fig. 5A). However, the expression levels of *TaFBN* genes varied among the different tissues. The expression levels of the *TaFBN*s in the tissues with high chlorophyll contents (leaf, shoot, and coleoptile) were significantly higher than those in other tissues. As shown in Fig. 5B, the expression levels of *TaFBN* were notably different at different developmental stages. The genes *TaFBN-A1*, *TaFBN-B1*, *TaFBN-A2*, *TaFBN-B2*, *TaFBN-D2*, *TaFBN-A3*, *TaFBN-A6*, *TaFBN-B6*, and *TaFBN-D6* were highly expressed at all developmental stages. However, the expression levels of *TaFBN-B4*, *TaFBN-D5*, *TaFBN-A9*, *TaFBN-B9*, and *TaFBN-D9* were inhibited at all developmental stages. The expression levels of other *TaFBN* genes did not change significantly during any developmental stage. These data indicated that *TaFBN* genes have tissue-specific expression patterns, and some *TaFBN* genes play a vital role in the growth and development of *T. aestivum*.

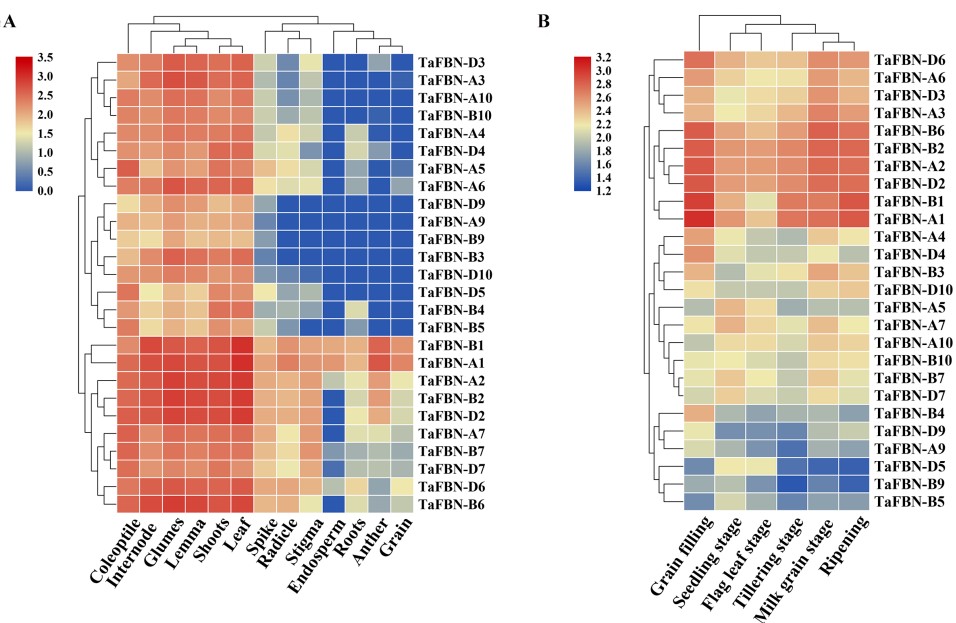

**Figure 5** **The expression of *TaFBNs* in various tissues and developmental stages.** (A) Tissue-specific expression of the *TaFBN* gene family in different wheat tissues; (B) the expression pattern of the *TaFBN* gene family at different developmental stages. A heatmap was created in TBtools software and based on the expression data. The color scale represents relative expression levels, with red indicating higher levels of expression and blue indicating lower expression levels.

## Expression profiles of *TaFBN* genes in response to abiotic stresses

To further clarify the potential functions of *TaFBN* genes under abiotic stress, the expression levels of *TaFBN* genes were analyzed under drought, stripe rust, cold and heat conditions. Most of the *TaFBN* genes were shown to be involved in the response to one or more abiotic stresses (Fig. 6). The transcripts of *TaFBN-A1*, *TaFBN-B1*, *TaFBN-A2*, *TaFBN-B2*, *TaFBN-D2*, *TaFBN-D6* and *TaFBN-B6* were significantly upregulated by drought, stripe rust, cold, and heat treatments. However, the expression levels of *TaFBN-A5*, *TaFBN-B5*, *TaFBN-D5*, *TaFBN-A9*, *TaFBN-B9*, *TaFBN-D9*, *TaFBN-A10*, *TaFBN-B10*, and *TaFBN-D10* were slightly downregulated under drought, stripe rust and heat stresses. In addition, most of the *TaFBN* genes were upregulated after 12 h of drought treatment and 11 d of stripe rust infection. Interestingly, almost all *TaFBN* genes had a high level of expression under cold stress. Other *TaFBNs* were induced to express under some of the stress conditions. The transcription levels of the tested *TaFBN* genes were significantly downregulated under drought stress conditions. These results indicated that *TaFBN* genes might participate in response to abiotic stresses, especially drought, stripe rust, cold and heat stress in *T. aestivum*.

## Validation of *TaFBN*s by qRT-PCR

To further detect the expression levels of the *TaFBN* genes in different tissues, we selected nine representative genes from the *TaFBN* gene family (*TaFBN-A1*, *TaFBN-B1*, *TaFBN-A2*, *TaFBN-B2*, *TaFBN-D2*, *TaFBN-B5*, *TaFBN-B6*, *TaFBN-A9*, *TaFBN-B9*, and *TaFBN-D9*)
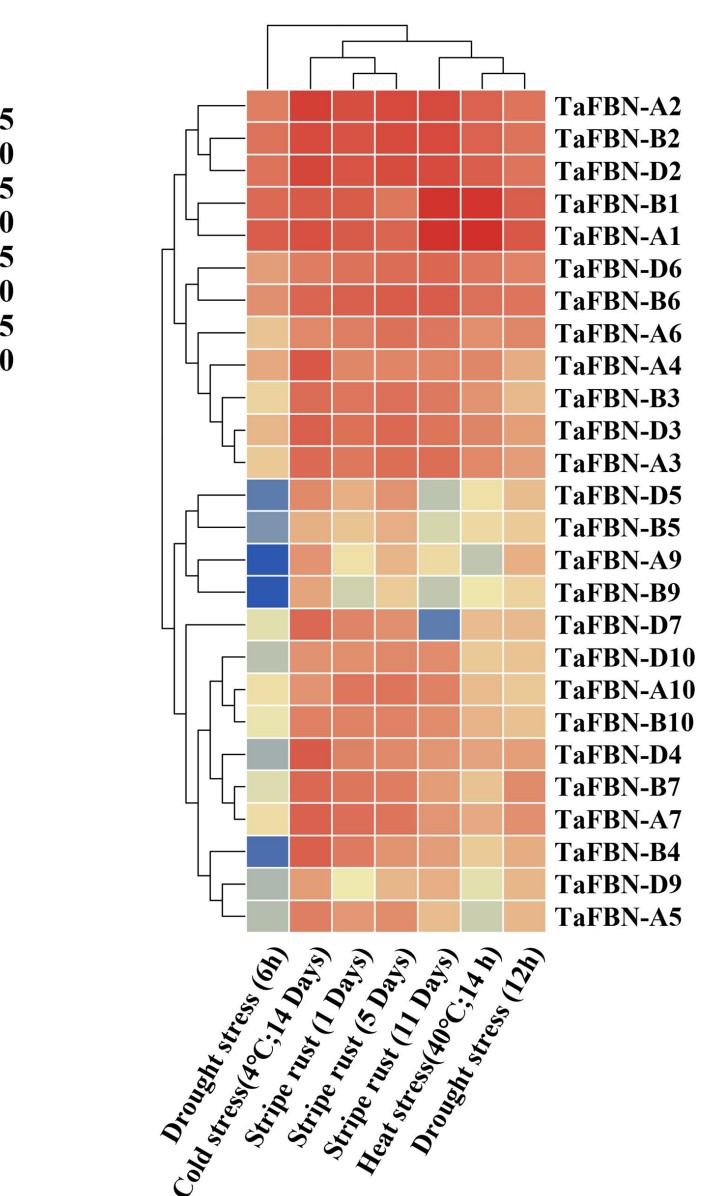

**Figure 6** **The heat map of expression profiles of *TaFBNs* in *Triticum aestivum* under biotic and abiotic stress.** Expression levels are indicated in different colors, with red indicating higher expression levels and blue indicating lower expression levels.

based on their expression profile, and analyzed their expression levels using qRT-PCR (Fig. 7A). The results showed that the expression of nine *TaFBN*s in the leaves and shoots was significantly higher than that in the roots. We also analyzed the *TaFBN* gene expression in leaves under drought stress in *T. aestivum* seedlings (Fig. 7B). The results suggested that the expressions of some *TaFBN* genes, such as *TaFBN-A1*, *TaFBN-B1*, *TaFBN-A2*, *TaFBN-B2*, *TaFBN-D2* and *TaFBN-B6*, were induced at different time points under drought stress. However, *TaFBN-B5*, *TaFBN-A9*, *TaFBN-B9*, *TaFBN-D9* displayed downregulation after

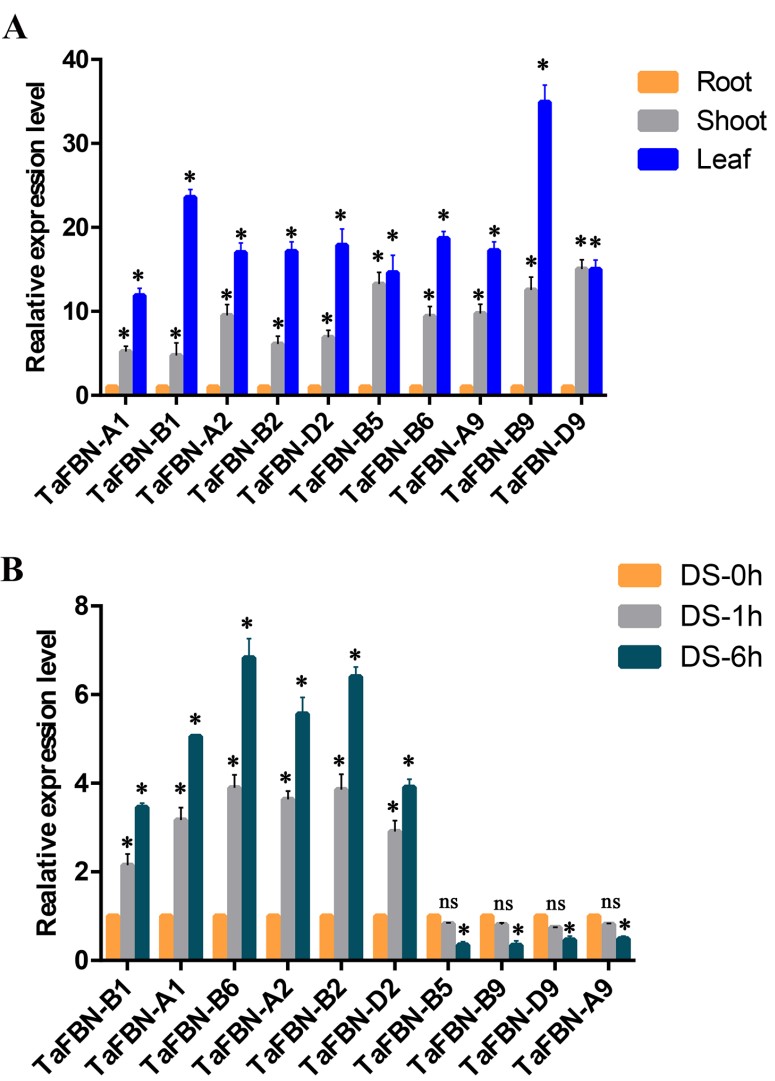

**Figure 7** **The expression analysis of *TaFBN* genes in different tissues and under drought stress using qRT-PCR.** (A) The relative expression levels of *TaFBN* genes in different tissues. (B) The relative expression levels of *TaFBN* genes in leaves after drought treatment for 1 h and 6 h. Each treatment contains three biological replicates.

drought treatment. In addition, as the treatment time increased, the expression level was significantly upregulated or downregulated. These results are consistent with the data of the above expression profiles.

## DISCUSSION

In this study, we identified 26 *FBN* genes in the *T. aestivum* genome. These genes were distributed on 11 chromosomes and had molecular masses ranging from 23.75 to 43.59 kDa and pI values ranging from 4.59 to 9.61. This diversity suggested that *TaFBN* genes may have specific biological functions in different metabolic processes. Furthermore,
the results indicated that most of the *TaFBN* genes were located on the chloroplast and contained chloroplast transit peptides. This provided strong evidence that various FBNs might participate in photosynthesis. The overall hydrophobicity of the protein sequences were calculated with GRAVY, with higher positive GRAVY values indicating a greater level of hydrophobicity (*Faya, Penkler & Tastan Bishop, 2015*). Almost all of the *TaFBN* genes' GRAVY values were negative, which meant that most of the proteins were hydrophilic. In contrast, previous studies have reported that the *FBN* family can bind to and transport small hydrophobic molecules in *A. thaliana* (*Singh & McNellis, 2011*; *Kim, Lee & Kim, 2015*). However, the specific spatial structure and the percentage of hydrophobic residues may affect the hydrophobicity of proteins (*Dyson et al., 2004*). Therefore, these different results may reflect the biological function diversity of the *TaFBN* genes.

To analyze the evolutionary relationships of the *FBN* genes, we constructed a phylogenetic tree with 183 *FBN*s from *T. aestivum*, *O. sativa*, *S. bicolor*, *Z. mays*, *P. hallii*, *P. virgatum*, *S. italica*, *H. vulgare*, *A. thaliana*, *B. oleracea* var. *capitata*, *N. tabacum*, *G. max*, and *C. arabica*. These *FBN* genes were divided into 11 subfamilies using the classification method described for *FBN* in *A. thaliana* (*Singh and McNellis., 2011*). In addition, gene family members of *TaFBN* were always clustered together with monocots in general, such as *H. vulgare*, *O. sativa*, *Z. mays* and *S. bicolor*. The similar exon-intron structures and the number of conserved motifs were observed in the same subgroups. These results suggested that the *FBN* genes located in the same branch may have similar biological functions in these monocots. At the same time, the evolutionary analysis provides a solid foundation for further functional studies on *FBN* genes in wheat.

Gene expression levels in different tissues and at different developmental stages may be determined by gene function. Previous studies have shown that *FBNs* are regulated by a variety of biological and environmental factors at different growth and developmental stages (*Singh & McNellis, 2011*). We analyzed the expression patterns of the *TaFBN* gene family in *T. aestivum* during different growth and development stages, and under biotic and abiotic stresses through publicly available gene expression data. We obtained 26 *TaFBN* gene expression profiles, which showed that most of the genes were highly expressed in the leaf, shoot, and coleoptile. Similar results have been reported in potato, *Arabidopsis*, and *Brassica rapa* (*Monte, Ludevid & Prat, 1999*; *Kim et al., 2001*; *Yang et al., 2006*). Furthermore, the expression profile data suggested that *TaFBN-A1*, *TaFBN-B1*, *TaFBN-A2*, *TaFBN-B2*, *TaFBN-D2*, and *TaFBN-B6* expressions were strongly induced under drought, stripe rust, cold, and heat stresses, but *TaFBN-A5*, *TaFBN-B5*, *TaFBN-D5*, *TaFBN-A9*, *TaFBN-B9*, *TaFBN-D9*, *TaFBN-A10*, *TaFBN-B10*, and *TaFBN-D10* expressions were slightly inhibited under these stresses. In addition, other *TaFBN*s responded to one or more stresses. These was some evidence that the level of FBN proteins, such as *FBN1a*, *FBN1b* and *FBN2*, increases in the leaves of rice, *Arabidopsis*, *Brassica* and potato plants subjected to drought and cold stress (*Gillet et al., 2001*; *Kim et al., 2001*; *Laizet et al., 2004*; *Lee et al., 2007*). Furthermore, the accumulation of the FBN1 protein in the tomato *flacca* mutant, which is defective in ABA biosynthesis, was significantly reduced compared to the wild type during drought stress (*Gillet et al., 2001*). We obtained similar results in wheat plants subjected to drought stress. It is possible that *FBN* gene expression was regulated through

endogenous ABA concentrations in response to numerous stresses (*Singh & McNellis, 2011*). Transcription factors participate in various biological processes by regulating the expression of downstream gene *cis*-regulatory elements (*Ning, Liu & Kang, 2017*). In this study, many *cis*-regulatory elements were detected in the promoter sequences of the *TaFBN* genes. These elements contained light response-related elements, drought response-related elements, and hormone response elements, such as MeJA, abscisic acid, gibberellic acid, salicylic acid, and auxin. Interestingly, all *TaFBN* genes included many light response-related elements. For example, *Rey et al. (2000)* found that overexpressing *FBN1* can promote plant height and flowering under high light levels in tobacco (*Rey et al., 2000*; *Singh & McNellis, 2011*). *Leitner-Dagan et al. (2006)* showed that *FBN* gene expression and carotenoid accumulation in the flower tissue of the cucumber increased during GA treatment, and GA-responsive elements were found on the FBN promoter sequences (*Leitner-Dagan et al., 2006*). By contrast, auxin (IAA) can delay the accumulation of FBN protein in bell pepper fruit, but abscisic acid (ABA) can promote this process (*Deruère et al., 1994*; *Singh & McNellis, 2011*). Although the expression patterns of *TaFBN* genes were varied and complex, overall, these genes had similar functions in plant stress resistance and chromoplast development (*Singh & McNellis, 2011*).

## CONCLUSION

In this study, we identified 26 *FBN* genes in *T. aestivum* using a genome-wide screening approach. Based on their phylogenetic relationships, these *FBN* genes were classified into 11 subfamilies. The *TaFBN* gene structures and conserved motifs were highly conserved in the same subgroup. Many *cis*-regulatory elements were found in the *TaFBN* gene promoter sequences, which showed that the expression of *TaFBN* genes was regulated by various hormones and environmental factors. Moreover, almost all *TaFBN* genxpression profies were highly expressed in the leaf, shoot, and coleoptile. The expression profiling data suggest that *TaFBN-A1*, *TaFBN-B1*, *TaFBN-A2*, *TaFBN-B2*, *TaFBN-D2*, and *TaFBN-B6* were responsive to many biotic and abiotic stresses. These results can help us to clarify the structural and functional relationships among *TaFBN* gene family members.

### Funding

This study was supported by the National Natural Science Foundation of China (No. 31560547), the Talents of Guizhou Science and Technology Cooperation Platform ([2017]5603), the Scientific and Technological Innovation Platform of Guizhou Province (No. 2014-4003), and the Research institutions Enterprise Action Program of Guizhou Province (No. 2014-4007). The funders had no role in study design, data collection and analysis, decision to publish, or preparation of the manuscript.

### Grant Disclosures

The following grant information was disclosed by the authors:

National Natural Science Foundation of China: 31560547.
Talents of Guizhou Science and Technology Cooperation Platform: [2017]5603.
Scientific and Technological Innovation Platform of Guizhou Province: 2014-4003.
Research institutions Enterprise Action Program of Guizhou Province: 2014-4007.

## Competing Interests

The authors declare there are no competing interests.

## Author Contributions

- Yaoyao Jiang and Haichao Hu performed the experiments, analyzed the data, prepared figures and/or tables, and approved the final draft.
- Yuhua Ma analyzed the data, prepared figures and/or tables, authored or reviewed drafts of the paper, and approved the final draft.
- Junliang Zhou conceived and designed the experiments, analyzed the data, authored or reviewed drafts of the paper, and approved the final draft.

## Data Availability

The raw data are available in the Supplemental Files and Fig. 1.

## Supplemental Information

Supplemental information for this article can be found online at http://dx.doi.org/10.7717/peerj.9225#supplemental-information.

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
