# Peer review of "Genome-wide identification and characterization of the fibrillin gene family in Triticum aestivum"

_PeerJ, doi:10.7717/peerj.9225_

## Round 0.1 · original submission · Major Revisions

All 3 reviewers agree that the manuscript requires major revisions before it could be considered as suitable for publication. Please provide a revised version taking into account all suggestions of the reviewers and an accompanying letter describing all changes. Please, note the annotated manuscript attached by Reviewer 3.

Furthermore revision of spelling and grammar errors is required.

Abstract: line 16:...family of genes....
line 19: rephrase...'less progress in molecular mechanism."

Reviewer 1 ·

Basic reporting

no comment

Experimental design

no comment

Validity of the findings

no comment

Additional comments

In this manuscript, using the reported FBN protein sequences in the wheat database, the authors identified 26 FBN genes and analyzed their biophysical properties and expression profiles of 3 TaFBN genes in different tissues and at different developmental stages and in response to abiotic and biotic stresses using the GENEVESTIGATOR database, the results are informative. Overall, throughout the manuscript, the data used all came from database, there is no own experimental design, it`s simple, so the following problems must be solved.
1. The FBN sequences for blast search should not be limited in only two species (Arabidopsis and wheat), more FBN sequences from other monocot plants should be added.
2. The more detailed version of obtained wheat genome sequences should be noted.
3. Line 202 mentioned ‘Gene structure analyses indicated that homologous genes had similar exon-intron distribution pattern’, but homologous genes, such as TaFBN2 and TaFBN10, TaFBN13 and TaFBN15, TaFBN6 and TaFBN21, TaFBN18 and 3, TaFBN17 and 25, TaFBN14 and 19, TaFBN22 and 26, TaFBN16 and 5, TaFBN12 and 20, seems have different gene structure.
4. According to the ‘Fibrillin is a highly conserved family of gene’ in Abstract, the sequences alignments of conserved domain should be exhibited.
5. The analysis results of TaFBN cis-regulatory elements, such as the indicated cis-regulatory element of TaFBN4, 5 and 6, should be associated with the results of expression profiles response to abiotic stress. If the two results were different, please explain the reasons in the discussion part.
6. Line 255 described that at least 25 pairs of TaFBN genes underwent gene duplication arising from segmental duplications. However, the 25 pairs of TaFBNs were genetically replicated by tandem replication in Abstract. Please be consistent.
7. Expression profile data used was 2015 and 2016 in which sequencing information of wheat genome was not newest and incomplete. So expression profiles of TaFBN families in different organs and stresses through RNA-Seq need to be analyzed based on the newest wheat database (2018).
8. As get the results of expression profiles of RNA-Seq, validate it through qRT-PCR assay.
9. The paper is full of grammatical errors, many discriptions are inaccuracy.

·

Basic reporting

In this study, Jiang and colleagues identified and analyzed the fibrillin gene family in wheat. The study is based on publicly available genomic and expression data. Although it could be of interest to better understand the evolution and molecular mechanisms of those proteins apparently participating in photosynthesis and involved in biotic and abiotic stresses, the authors failed to correctly interpret some of their results.
Moreover, there are multiple typos (e.g. line 261 “Tiussse”) and bad English formulations (e.g. line 257 “could be driven the evolution process”) all across the manuscript. Some paragraphs are relatively well written but others were not carefully edited. Therefore I would recommend the authors to get it proofed by an English speaking colleague or, at the very least, using an automatic spelling software (either the one directly provided in Microsoft Word or the one by Grammarly which performs pretty well, for example).

Availability of data: alignments of FBN genes/proteins in wheat and across the other species presented in Fig. 3 should be provided as fasta (or equivalent) format.

Citations are not always in the correct format: “Multiple references to the same item should be separated with a semicolon (;) and ordered chronologically.”

Experimental design

Phylogenetic trees must have bootstrap values like in Figure 4. Why showing three times the same tree but indicating only in the last one the support values?

Also, the references and accession numbers of all the sequences used should be mentioned.

Why not using the Wheat Expression Browser (http://www.wheat-expression.com/) to explore the expression patterns of TaFBN genes? All the genes are available and not only three.

Validity of the findings

The main flaw of this study is that the authors confused gene duplications and merging of genomes. The authors are apparently aware that wheat is a hexaploid species but they seem to ignore the mode of formation of this species and its implication for homologous genes. Indeed, they consider homeologs as duplications although they were brought together following inter-species hybridization and allopolyploidization. Those gene copies were completely homologous in the ancestral species. This is to distinguish from duplication as it can be seen in Arabidopsis or Zea mays, species that are diploids but known paleopolyploids and present sometimes a duplicated fibrillin gene. It is still possible that there are duplications especially on the chromosomes 2 but Figure 3 seems to indicate, if the node support values are meaningful, that the duplications are much older and shared by all the species included in the tree.
This also means that the values obtained in Table 2 do not reflect a duplication date but maybe the date of divergence of the progenitors of wheat. Also, I would have to criticize the method used. Indeed, the authors used a method and a value drawn from Koch et al. (2000) although this very same article describes and discusses different values of the synonymous substitution rate among which one that was estimated from grasses. Another article of high-interest discussing the uncertainty associated with those values is Senchina et al 2003.

Additional comments

Homeologs should be identified with their genome location (A, B or D) for example TaFBN22 and TaFBN26 could be named, due to their clustering with AtFBN1a and AtFBN1b in the Group 1, TaFBN-B1 and TaFBN-A1, respectively. The way the authors named the genes looks random and does not facilitate the interpretation of the results.

Reviewer 3 ·

Basic reporting

In terms of form, I did not find the figure captions.

Experimental design

This paper is an in silico analysis (there is no experiments) aiming at identifying Fibrillin genes in Triticum aestivum and studying their potential functions based on their expression, their structure, the prediction of their physical and chemical properties and ci-elements in their promoters.
As far as this family has not been studied in wheat, the paper is interesting but it has to be thoroughly modified, particularly by using the last version of the wheat genome to identify TaFBN genes (https://www.wheatgenome.org/Tools-and-Resources) and their expression (http://www.wheat-expression.com/). Of course, data extracted need to be treated to express the DEG according to the different conditions.

Validity of the findings

no comment

Additional comments

There are mistakes all along the text that I underlined in yellow. Also, I added comments throughout the text.

Introduction
The introduction needs to be improved in terms of the roles of fibrillins. Is there any review?
The overall introduction needs to be read carefully mistakes all along the text (ex: line 257 “could be driven BY”, line 261 “tissue “, etc…

Materials and methods
Did you check genevestigator results in real? It would have been great to run qRT-PCR on selected genes to validate the results.
Did you identify homeologous genes? In other terms, in your 26 TaFBN, do you have triplets? Can you precise?
You would need to use the last version of the wheat genome (https://www.wheatgenome.org/)
For gene expression, you may use www.wheat-expression.com platform (Borrill, P.; Ramirez-Gonzalez, R.; Uauy, C. expVIP: A Customizable RNA-seq Data Analysis and Visualization
Platform. Plant Physiol. 2016, 170, 2172–2186.), to retrieve more information

Results
Why did you analyze the expression of only 3 TaFBN genes? Moreover, you can not tell that line 322, that “you analyzed the expression patterns of TaFBN gene family”.

Conclusion
The last sentence is too global. I recommend to remove it.

Annotated reviews are not available for download in order to protect the identity of reviewers who chose to remain anonymous.

---

## Round 0.2 · Minor Revisions

The revised manuscript has improved, but there are still some issues raised by the reviewers. Please, address those in a further revision including an accompanying letter.

Please, make sure that all changes are formulated in correct English language and note the annotated manuscript attached.

Reviewer 1 ·

Basic reporting

no comment

Experimental design

no comment

Validity of the findings

no comment

Additional comments

Thank you for revising the manuscript. It looks better now, especially in the genome-wide identification and classification parts. There are still some suggestions for your responses and supplementary experiments.
1. The identification and homology comparison in these five monocot species should be compared in detail in the discussion. The analysis of gene structure, function domain and motif distribution of 26 TaFBN genes were good jobs, but the gene function from previous studies among different groups and different species should be introduced. This is important to the following analysis about cis-regulatory and gene expression pattern.
2. The cultivation condition should be more concrete, for example, the soaked “room temperature” in line 118 and germinated temperature should be a certain value. Some typos like the word “NaOCl” in Line117should be checked and revised.
3. The additional qPCR experiments were used to validate the expression profiles of RNA-Seq. The results in Figure 8a should correspond to the tissue special expression patterns of TaFBNs in Figure 6a. The results in Figure 8b, too, should correspond to the expression profile data downloaded from the website. If there is a deviation between website and experiment data, please explain the difference in the discussion.
4. The difference between PEG6000 and drought stress treatment methods should be listed in line 296. The obvious difference in expression profile of 26 TaFBNs between PEG6000 and drought stress should be explained in discussion. Based on the heatmap in Figure 7, the TaFBNs are more involved in cold stress and stripe rust stress. Please explain why you only chose drought stress for the validation of expression profile data.
5. The data of qPCR in Figure 8 were from three biological replicates in line 196, but the standard errors in figure were marked incorrectly, for example the standard errors in “DS-0h” should be 0.

·

Basic reporting

In general the authors amended their manuscript with the recommendations of the reviewers.

However, although the authors mentioned that the text was carefully corrected by professional language company, I still think that some formulations could be omitted ("It must be fortunate that", "As we know"...). The authors can find my suggestions in the annotated manuscript. Some references are missing (wheat genome sequence)

Experimental design

It is good to compute the bootstrap but it is also important to show it! On Figure 3 please indicate the bootstrap support at least for the main clades, a tree without bootstrap values is meaningless!
The authors should provide the accession numbers of all sequences including the non-wheat sequences (Arabidopsis, Zea...)

Validity of the findings

Line 209, the authors find it interesting that all the sequences that they analyze contain a conserved PAP_FBN domain but one of their selection criteria was that to be considered in their dataset, a protein should have a PAP_FBN domain! Maybe that's what is in the Sup. figure S1 and S2 but the captions are missing.

There is still a paragraph misinterpreting the results ll. 313-317 with regards to the polyploidy of wheat and the number of genes identified. Here again you can find the full comment annotated on the manuscript.

---

## Round 0.3 · Minor Revisions

The manuscript has greatly improved, but there are still some points of the reviewers. Please, recheck format of the references and note the annotated manuscript file in the attachment.

Please, revise the section line 326-334. The reason that there are higher numbers of FBN genes in wheat is most likely due to polyploidy. Also it is questionable if general evolutionary conclusions can be based on just one gene family. In figure 3 there is also clustering of H. vulgare FBN genes with wheat FBN genes found.

Please, provide the revised version of the manuscript and accompanying letter.

Reviewer 1 ·

Basic reporting

no

Experimental design

no

Validity of the findings

no

Additional comments

Thank you for the positive response to the previous suggestions. In this article, fibrillin (FBN) family genes were identified and characterized in wheat based on the whole genome of T. aestivum. The expression pattern of TaFBNs in different tissues, developmental stages and abiotic stresses were analyzed and verified by qRT-PCR experiments. It was useful for the research on the wheat resistance. After some modification, the article is better than before. More analyses of results data have been added in the introduction and discussion, and some words and sentences in language and figure format have been corrected. The inconsistent format in references are still obvious, for example, the absence of DOI, the italics of the journal names, the inconsistent paragraph format and so on, check and correct.

·

Basic reporting

Manuscript amended according to the reviewers' suggestions. However, in the newly added parts the English needs some correction. I annotated the PDF wherever I spotted some problems, but again professional English editing would be necessary.
The reference section is weirdly formatted with words and numbers cut in between lines without dashes (for words).

Experimental design

The phylogenetic trees are now correctly displayed with support values but it is good practice to collapse low support nodes (< 50) and to use a symbol (*) for high support nodes (> 99). This should make the trees less crowded. For collapsing the nodes, if the authors don't know how to do it, I would recommend Newick utilities (Junier and Zdobnov 2010; http://cegg.unige.ch/newick_utils).

Validity of the findings

The interpretation of the phylogenetic results is just wrong in the way it was recently amended (please see the annotations on the manuscript). The authors should first check the literature, there is a large amount of studies on phylogenetic relationships, especially between crop species. The close relationship of barley and wheat is known for long and yet, the authors speculate that rice is closer to wheat based on this one gene family and their wrong interpretation of the results while this particular question is not the focus of their study.

Additional comments

Please consider taking on-board a phylogeneticist if you are planning to use evolutionary relationships in your work.

---

## Round 0.4 · Minor Revisions

Unfortunately, the current version of the manuscript still contains a number of errors and typos. Please, carefully correct the manuscript according to the reviewer's suggestions and please note the annotated version of the manuscript in the attachment with many useful and necessary improvements.

Please, provide a corrected final version of the manuscript.

·

Basic reporting

There was still some weird sentence structure and typos. I tried again to annotate on the manuscript the ones that I spotted but I ask the authors to also take care to clean up the maximum of typos, after a third revision I do not expect to see a "The" in the middle of a sentence (see line 91).

Experimental design

The authors made some changes on the Figure 3 as I recommended in the previous revision and reported in the rebuttal letter the method that they used but failed to add these modifications in the methods part and in the caption. Sorry, if I was not clearer previous time but I meant to replace bootstrap values > 99 with an asterisk. Also, I am not sure why the authors suddenly decided to make a phylogram (branch length relative to the amount of changes) instead of the cladogram (branch length meaningless) that they were showing earlier, this should also be possible with a consensus tree with collapsed nodes.

Validity of the findings

Please remove ALL mentions of duplication in wheat. The duplications that the authors are describing are not specific to wheat but common to all plant organisms (at least the ones on the figure 3), the only thing to mention would be that duplications happened a very long time ago. What about much more distant organisms (mosses, algae...).
Actually, the only peculiar thing for wheat is the loss of the FBN group 1 from the D genome, all the rest is in accord with the phylogeny (remember that wheat is hexaploid, so we expect at least three homeologs for each single-copy gene in a diploid species). Among the species included in the figure 3, many species are apparently showing a much more interesting pattern: duplication of group 7 in A. thaliana, duplication of group 3 genes in most dicotyledons, N. tabacum for group 2... but I am aware that it is not the focus of this study.

Additional comments

I strongly recommend the authors to correct all the annotations that I made. I realized that the authors did not necessarily understand the simple annotations so this time I took the time to provide them with full correction. The fact that some simple annotations were not considered by the authors was very upsetting!

---

## Round 0.5 · accepted · Accept

The current version of the manuscript can be accepted for publication in PeerJ.